# Video based monitoring systems for hand hygiene compliance auditing: What do patients think?

Katherine J. McKay[1,2]*, Ramon Z. Shaban[1,2,3,4]

**1** Susan Wakil School of Nursing and Midwifery, Faculty of Medicine and Health, University of Sydney, Camperdown, NSW, Australia, **2** Sydney Institute for Infectious Diseases, Faculty of Medicine and Health, University of Sydney, Camperdown, NSW, Australia, **3** New South Wales Biocontainment Centre, Western Sydney Local Health District and New South Wales Health, Camperdown, NSW, Australia, **4** Communicable Disease Branch, Public Health Unit, Centre for Population Health, Western Sydney Local Health District, Westmead, North Parramatta, NSW, Australia

\* Kmck4637@uni.sydney.edu.au

**Data Availability Statement:** All relevant data are within the manuscript and its Supporting Information files.

**Funding:** The study was supported by a financial grant from the Marie Bashir Institute for Infectious

## Abstract

### Background

Compliance with hand hygiene by healthcare workers is a vital aspect of the quality and safety in healthcare. The current method of monitoring compliance, known as direct observation, has been questioned as have the various electronic measures proposed as alternatives. In our earlier work we established the capacity of video-based monitoring systems (VMS) to collect data with increased efficacy, efficiency and accuracy. However, the spectre of the approach being seen as an unacceptable invasion of patient privacy, was raised as a barrier to implementation by healthcare workers.

### Methods

In depth, semi structured interviews were conducted with 8 patients in order to explore their beliefs and options regarding the proposed approach. Interviews were transcribed and then thematic and content analysis was conducted in order to uncover themes from the data.

### Results

Despite healthcare worker predictions, patients were generally accepting of the use of video-based monitoring systems for the auditing of hand hygiene compliance. However, this acceptance was conditional. Four interconnected themes emerged from the interview data; quality and safety of care versus privacy, consumer Involvement–knowledge, understanding and consent, technical features of the system, and rules of operation.

### Conclusion

The use of within zone VMS approaches to hand hygiene auditing has the potential to improve the efficacy, efficiency and accuracy of hand hygiene auditing and hence the safety and quality of healthcare. By combining a suite of technical and operational specifications

Diseases and Biosecurity, University of Sydney of $15,000AUD. No other financial or material support was received, including from the authors' institutions. The contributing funder had no role in the study design, data collection and analysis, decision to publish, or preparation of the manuscript whatsoever. The authors do not receive a salary from the contributing funder.

**Competing interests:** The authors have declared that no competing interests exist.

with high level consumer engagement and information the acceptability of the approach for patients may be significantly enhanced.

## Introduction

Hand hygiene compliance by healthcare workers (HCWs) and the measurement therein is unquestionably important. The WHO "5 Moments for Hand Hygiene" provide a framework for the performance of hand hygiene such that the interruption to microbial transmission is optimized the details of which is outlined in the "WHO Guidelines on Hand Hygiene in Healthcare" [1]. Direct observation is held up as the premier data collection method, yet while having many advantages, is also increasingly being questioned as flawed. These issues principally relate to the time and hence cost of data collection which is substantial [2–6]. Added to this is an inherent inaccuracy of any data collected via direct observation due to the biases that a human auditor invariably brings to the process of data collection. Various electronic measures have been proposed as an alternative way to collect compliance information, the major issue with their use is however, the reliance of proxy measures of compliance rather than the WHO 5 moment criteria [7–9]. This gap led us to explore the potential to use video-based monitoring (VMS) approaches to collect data based on the rational that by combining direct observation and vision-based surveillance approaches data would be able to be collected and behaviour change effected with greater efficacy, efficiency and accuracy. Throughout the process of developing and examining the use of VMS to record footage and subsequently audit hand hygiene compliance, the issue of acceptability to both HCWs and patients remained prominent. And while we demonstrated the ability to collect and audit hand hygiene behavior not only according to the WHO 5 moments criteria but with significantly improved effectiveness, proficiency and precision, the question of whether the palatability of the approach would prove a barrier remained [10]. While the HCWs who took part in the interviews which formed our earlier studies indicated that they could see the benefits of a VMS approach to hand hygiene auditing, the first disadvantage that they invariably raised was that the use of such technology would not be tolerable to patients who would see the use of the required within patient zone cameras as an unacceptable invasion of their privacy [8, 10, 11]. The term "patient zone" is derived from the 5 moments for hand hygiene framework and is seen as the space temporarily dedicated for the use of the patient–this may be an individual patient room, a curtained off section of a multi-bed room, a cubical or a treatment bay. In the context of VMS this does not include a patient bathroom or toilet. A review of the literature suggested a dichotomy, with some authors seeing the use of cameras within the patient zone as problematic, citing "inherent tensions" [12], "special challenges" [13], "serious issues" [14] and the matter being "fraught with problems" [15]. In contrast Clack, Scotoni [16] found that patient objections to filming were quite rare once information was provided. While a survey conducted by Raghavendra and Rex [17] found that 81% of patients undergoing a colonoscopy were interested in having their procedure recorded and 63% were prepared to pay for it. Diefenbacher, Sassenrath [18] speculate "overall privacy issues might be of low concern to both HCW and patients, at least when they are properly informed" (p. 501). However, it was also clear that the area required further investigation and it was this that prompted us to include the attitudes of patients in our data collection relating to acceptability. This study examined patients' perspectives of video-based monitoring systems for hand hygiene compliance auditing.

## Methods

### Study design

In-depth semi structured interviews within a pragmatic naturalistic framework were utilized in this study. Pragmatic naturalism is an approach which holds that genuine inquiry must be undertaken in a reliably empirical manner while also remain responsive to genuine human problems [19].

### Study setting and participants

As part of a larger study conducted in Australia and exploring the attitudes of key stake holders (hand hygiene auditors, frontline healthcare workers and consumers) we interviewed a convenience sample of 8 patients in order to examine their opinions as to the acceptability of the use of a VMS for auditing hand hygiene compliance. Initial recruitment was via email mediated by the Consumers Health Forum of Australia, interest was promulgated by word of mouth through associated networks and in total 8 patients volunteered to participate. Participants were predominantly female (75%) and most resided in Victoria, Australia (87%). They ranged in age from 19 to 85 years with an average age of 54 years. All had received inpatient care in a healthcare facility within the preceding 12 months, apart from one participant who was the parent of 2 children who had received inpatient care within the 12-month timeframe.

### Data collection and analysis

Following written informed consent signified by the return of a signed participant consent form (S1 File), semi-structured interviews were conducted with the 8 participants via Zoom™ by one researcher (KMc). Interviews lasted between 20 and 65 minutes and included questions about both current direct observation and the proposed VMS auditing methods. Participants were shown short videos in order to illustrate both direct observational and video-based auditing approaches (Fig 1). The individuals in these videos has given written informed consent to utilise the recorded footage for ongoing data collection and publication. All interviews were audio recorded with the consent of the participants. A question guide (S2 File) was used and was modified as the interviews progressed based upon concurrent analysis, which suggested new or expanded areas to explore with subsequent participants [20]. The use of open-ended questions facilitated rich discussion and the expansion of ideas and allowed concepts to emerge from the data. Recorded interviews were transcribed verbatim by the Researcher (KMc) which aided immersion and engagement [21] and member checking was undertaken to enhance reliability of the data, with transcripts being returned to participants to review as well as confirmation or refutation of ideas between participants to assist in establishing normative responses [22, 23]. The data from interviews underwent content and thematic analysis [24]. The semi-structured questions guided the use of open coding approach and allowed for the generation of categories which helped to describe and understand the data. Not only did this help to ensure the credibility of the data but also saw the development of relationships within the emerging themes [23, 25, 26].

### Ethical considerations

The study was conducted in accordance with the ethics approval granted by the Human Research Ethics Committee (HREC) of the University of Sydney (approval number 2021/040). "The individuals pictured in Fig 1 have provided written informed consent (as outlined in PLOS consent form) to publish their image alongside the manuscript.

| Video One | Video Two | Video Three | Video Four |
|---|---|---|---|
|  |  |  |  |
| Direct observational hand hygiene auditing taking place in the clinical setting | Footage of simulation scenarios captured by fixed, continuously recording cameras with front and side views | Footage of simulation scenarios captured by temporary, proximity activated cameras with rear views | Footage of simulation scenarios captured by temporary, proximity activated cameras with rear views with the addition of privacy filter (facial pixilation) |
| 10 seconds | 53 Seconds | 65 Seconds | 58 Seconds |

**Fig 1. Videos shown to participants to demonstrate various data collection methods for hand hygiene auditing.**

## Results

This examination of patients' perspectives of video-based monitoring systems for hand hygiene compliance auditing revealed 4 interconnected findings;

- Quality and safety of care versus privacy

- Consumer Involvement–knowledge, understanding and consent

- Technical features of the system

- Rules of operation

### Quality and safety vs privacy

Despite the suggestion that patients would object to the use of a VMS which involved within zone cameras, this did not prove to be the case for participants in this study. None indicated that concerns for their privacy was a barrier to the acceptibility of the approach.

> *"I think if it was done in the right way, it's not an invasion of privacy. Yeah." [PATIENT 3]*

Participants did indicate that a patient's room or bedspace was different to the use of cameras in pubic, which they aknowledged that they were both aware of and largely desientitised to.

> *"It's not something that I'm too bothered by, or think, too hard about, but I guess yeah they are around." [PATIENT 3]*

The word "vulnerable" was the one used most frequently in relation to the differences between CCTV and within zone recording,

*"It's the privacy and the vulnerability and the personal nature of that space that makes it more sensitive."* [PATIENT 8]

However notwithstanding concerns about the sensitive nature of within zone recording, most saw the approach as a form of "trade off" where the potential impact upon privacy needed to be weighed against patient saftey and improvements to the quality of care.

*"I guess it's a balancing of risk . . .. what is the risk profile of a particular area, what's the importance of knowing that specific procedures were followed? What is the danger, what is the consequence of them not being followed regularly."* [PATIENT 1]

All participants recognised the problems associated with the current direct observation auditing approach and that as a result the collection of representative data was not possible with that method;

*"Like they're sort of standing there right next to you, of course you're gonna wash your hands, so it doesn't exactly prove that it happens all the time. . . it just proves that it happens when someone's watching.* [PATIENT 5]

They also recognised that a VMS would allow data to be collected in a far more efficient, effective and accurate manner.

*"You would be able to actually view. . ..to look back and review things. . . so say a second time . . . not just have a personal bias for the one person. Yeah, with the cameras that turn on and off it means you know you're storing a lot less data, you can speed it up too. . .. And then obviously <u>actually</u> observing what people were doing it would help you measure and find out what's happening with. . .. If they are following that hand washing thing or any other practices not just hand hygiene "* [PATIENT 6]

While some participants indicated that they were very comfortable with the idea of within zone cameras, others were less certain, indicating that it would be some that they would have to become accustomed to over time.

*"Yeah. . . well I don't understand why anyone would have problem with it in the first place given that hygiene is imperative."* [PATIENT 5]

*"It would be very intimidating, like obviously I'd be able to get used to it but it's a bit jarring at first."* [PATIENT 6]"

Notwithstanding the oposing opinions regarding comfort with the approach all participants indicated that they would be accepting of a within zone VMS if they knew it was going to improve the quality and saftey of the care that they, and others, particularly those who were vulnerable, received.

*"I think if it meant that I was helping get better practices and making sure that everybody did the right thing, then I wouldn't mind."* [PATIENT 8]

*"Maybe for those with disabilities or compromised immune systems or those who are not verbal and can't speak for themselves. . .. if there is something monitoring what is happening*

*around them . . .that is more than likely a good thing. . . . . . . . . a security that there is monitoring." [PATIENT 1]*

Indeed, some participants saw the approach as not only acceptable, but necessary in order to optimise patient care;

*"This is what I come back to. . . . . . having gone through many years of life. . . .. I believe that if it's um. . . a. . . not a mandate, but if it's something that's necessary for the care of the resident. And let's face it, you know . . .. we would hope that infection control is controlled, and if this is part of doing that then I don't think it's an invasion at all, I believe any medical procedure that is done for the benefit of the patient, or resident is something that one accepts as part and parcel of the care. . . ..I'll be quite happy." [PATIENT 4]*

This 'acceptance' however was not unconditional, participants placed caveats on their willingness to embrace the use of a VMS as was emphysised by the use of phrasing such as "not if" and "so long as" indicating that patients needed certain assurances regarding the approach.

## Consumer involvement–knowledge, understanding and consent

Participants made it clear that they needed to know how the system would work, to understand why the data was needed and to thus be able to provide informed consent to the process. In terms of 'how it would work' participants indicated that they would want to be conversant not only as to the presence of cameras in their room, but also how and when they would opporate and what they would be recording.

*"But I think this is one of the things that should be raised explicitly . . .so that people don't end up . . . you know sitting in their room, they look up "Oh what's that camera doing up there, how long has it been there, when is it working and what is it recording?"" [PATIENT 1]*

*"I personally would like to know what the cameras are going to be used for. And if I was assured that they would only be for x y or z. . . and I. . . that was fine with me, then that. . . the whole thing would be fine with me. " [PATIENT 4]*

*"Just transparency when it comes to it, so obviously just up front. . . just talking about everything. . . .. Ummm. . .. And that's really it." [PATIENT 6]*

In addition to what was being recorded and when and how that would occur, participants also expressed the need for clarification with regards to the management and retention of the recorded footage.

*"What they do with the footage . . .. the security of the footage. I think is a really big issue. What happens with the footage. . . How long is it retained, who gets it, who keeps it, who owns it. That would be an issue I would think.. " [PATIENT 8]*

However it was not just the more practical aspects that participants felt needed to be explained, the focus also need to be on the 'why', that is, the purpose of the data collection as well as the benefits of the specific approach.

*"If patients understood why it was done and the benefits of doing it that way. . .. I think, if it was clearly explained to people, why it was done that way, and what the benefits were that most people would be okay with that." [PATIENT 3]*

However as several participants highlighted, there may be a lack of understanding among the 'general public' as to the importance of compliance with hand hygiene and other infection prevention behaviours.

*"Well I believe . . . I feel it would be safer. But I think for people who have not been involved, in medical things . . .. They do need to have every, every bit of conversation, explained, and so that . . . ah . . .. anyone who's not familiar, would have as much explained to them as possible."* [PATIENT 4]

The following response by PATIENT 5 illusrates the lack of awareness that many patients may have regarding the importance of hand hygiene and other infection control practices.

*"But you don't even think about hygiene . . .while you're there and it it's be good to know that's being worked out behind. . .. without you thinking about it. . .yeah. . . that someone's looking at it because I think . . ..Yeah,. . . so. . .. that's not something that I've ever really paid too much attention to. . .*[PATIENT 5]

As well as understanding the purpose of recording per see, participants also felt that they would want the assurace that the data was being used for not only the reasons that they had been told of, but also that these purposes were 'good' or beneficial.

*"I guess just like not misuse of it . . . like a guarantee of that . . . and then it is actually being used for the reason stated.. in an agreement or something, that's about it"* [PATIENT 6]

Notwithstanding having gained an understanding or the how and why of the recording process and despite seven out of eight participants indicating that they favoured an "opt out" approach where recording was the default and the onus was on the patient to decline, the importance of consent was clear. Only one participant felt that 1:1 written concent was required. All others suggested that patients could be informed upon admission of the use of VMS within the facility and/or could be provided with. . .

*". . .. just some open information to say this is what we'd be doing. And if you've got a problem with it, just to approach whoever they can approach to say, "Listen, I'm not comfortable with this." . . . you know"* [PATIENT 5]

PATIENT 1 was more explicit in their recommendations as to the process, suggesting simple written information as well as translations for non-english speaking patients.

*"Indeed I would recommend using . . . you know . . . an easy read sort of format where you know you've got pictures, pictures of cameras and staff, you've got pictures of patients doing things so there could be no or at least minimal risk of people misinterpreting. This is a camera, this is what it does, and these are the times . . .. Dot . . . dot . . . dot when these things are happening, when we will be recording. And you should be able to put that . . . I think . . . in one relatively clear page with the question at the bottom . . . "do you consent to this happening?"* [PATIENT 1]

Implicit in the notion of concent was the concept of control. Participants indicated that it was important for them to know that they had control over the recording process and that they could opt out without penalty be it all together or in relation to a specific aspect of their care.

*"So yes, if somebody feels that it's a bridge to far they can say no, please turn that off….if . . . the patient doesn't want it to operate . . . then press the button and the cameras don't operate " [PATIENT 1]*

Participants also raised the possibility of the cameras of a VMS being put to other uses including patient monitoring for falls or behaviors of concern, staff or patient safety, telehealth consultation including the use of medical interpreters as well as family communication or meetings, the latter being particularly pertinent when visitors were unable to be physically present. While all participants were supportive of such uses for the technology, they were firm in their beliefs that specific consent would be required for each usage.

*"Yeah, I think it's worthwhile if you've got the resource there to have multiple purposes, obviously, individual consent for any of those.. that how it's going to be used, but I. …. but yeah. . . I think using it for multipurpose makes sense from a resource perspective…..you would have to get specific permission for each" [PATIENT 3]*

Althought all higligted the importance of consent of and of the patient having control over the recording process, 2 participants indicated scenarios where they felt that recording without consent should be permitted. This was in situations where staff or patient safety was at risk and the use of video-based monitoring would be of clear benfit.

*" Obviously, I mean, you'd have to look into the legalities of this, but you know. . . if a patient presents with physical signs of abuse. If…... if you do suspect that there's something going on…. if there's…. a reason to feel that that's the case, I think, yeah…. definitely, if there's a need….. If you feel that something's occurring, and that would obviously be written notes anyway that would.. that the recording's happening for a reason, then. . . if you can justify it…. sure …..fine." [PATIENT 5]*

Similarly, PATIENT 7 indicated that they felt video-based monitoring could serve the duel function of auditing and protecting patients and staff.

*"I do believe that . . . I think the type of patient that are coming in to hospital these days can be quite abusive, verbally and physically. . . you know drug induced behaviors. . . turn nasty at any time . . . I believe they should be able to be monitored at all times" [PATIENT 7]*

### Technical features of the system

Having been shown examples of camea based approaches, participants were able to identify the features (Fig 2) that they felt enhanced the palatabilty of the method. While individual aspects were important, particularly the proximity activation feature which all endorsed, it was also the entire bundle that was seen to be of benefit.

*"And I think it would be a package of those thing and look I would also recommend that if it was open to me" [PATIENT 1]*

*" I think all those things would give a pretty comprehensive suite of things to kind of make people more comfortable." [PATIENT 3]*

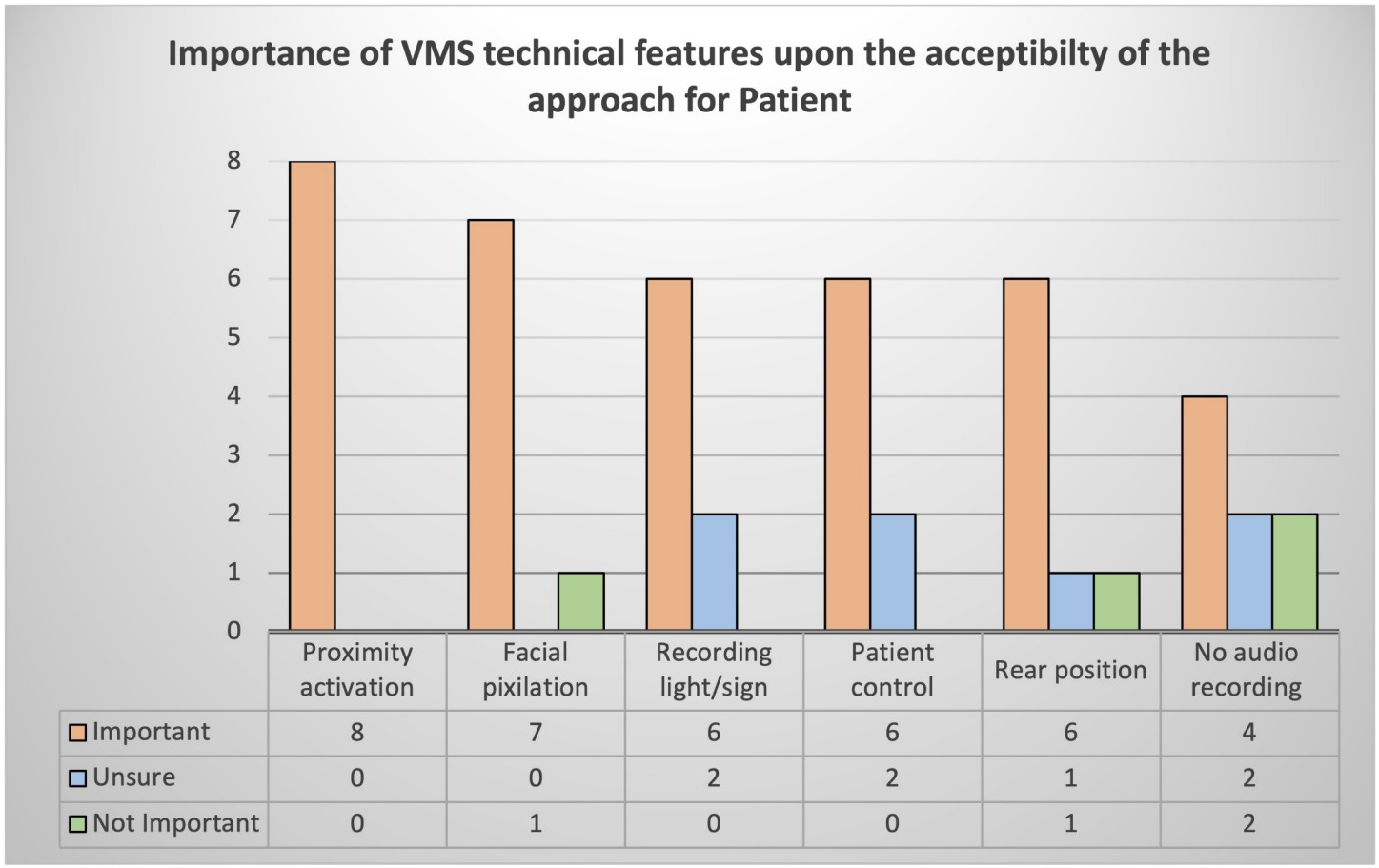

**Fig 2. Importance of VMS technical features upon acceptibilty of the approach for consumers.**

In particular, it was the proximity feature, where the camera will only record when the HCW is close-by, via Bluetooth beacon activation, which was seen as highly favourable. Participants indicated that knowing that they would not be recorded when alone in their room, sleeping for example, was a major "seller" of the approach, they also indicated that they found the use of proximity activation less intrusive and that the focus was "*more on the worker*" [PATIENT 3].

*"Yeah, I think that would improve, improve my acceptability of it. . .. if it only activated. If I could be 100% guaranteed and only activated . . . if. . . when the health care worker came into the room." [PATIENT 8]*

Conversely, several noted that they would feel a sense of reassurance that interactions with HCW could be captured. One participant shared a particularly harrowing experience;

*"Yeah, well I mean, in my instance, I wish they had of been cameras. Because, Yeah, yeah, I had a very bad, very bad experience with a nurse. . . extremely bad. . . . . ..Oh my god. . .. It was horrific . . . It was just.. the most horrific experience of my life, and I just lay in bed, I just cried. . .I knew but I knew her nursing was not up to scratch, but I . . .like I said was in a vulnerable position. . . and I thought I didn't have a right to say anything. . ..BUT. . . I don't think she'd have a job if she had a camera on her " [PATIENT 8]*

In a similar vein, PATIENT 5 explained how, as a parent, they would find the use of proximity activated cameras reassuring.

*"Like when you've got kids in hospital obviously you wouldn't want cameras on all the time because you're not always there, but if they're going on when just healthcare providers are there, that also is a bit of peace of mind as well there, because if you aren't in the room and a nurse walks in, and then it comes up and something's happened or hasn't happened then there's some proof. . . you know it works to look after, nurses and doctors as well because there's proof of what's occurred"* [PATIENT 5]

Other features that were seen as favorable were facial pixilation, which was seen as creating a sense of comfort through anonymity and what PATIENT 4 called "*double reassurance*" and rear camera position where the field of view was from above and behind the patient. The use of a light or illuminated sign to indicate recording was in progress was favored by 75% of respondents as was the ability for patients to deactivate or control the system. Lack of audio recording was not rated as highly as an important feature, although when the desire to avoid the risk of HCWs or patients self-censoring was explained most could see the advantages. Continuous recording, front on camera view and to a lesser degree, the ability for HCWs to control the cameras was seen as less or unacceptable.

*"I would HATE to be in a room where I was being filmed ALL the time.. as a patient. . . at your lowest point. I would hate it. Absolutely! . . . I'd hate. . . I'd hate to have that."* [PATIENT 8]

*"what if they (HCWs) turn it off to do something that they shouldn't be doing and then part of the evidence is not there.. and so the questions come in. So, that is a hard one. . . I don't know"* [PATIENT 5]

## Rules of operation

For participants, the concept of rules related to the way in which the footage was managed, in particular, they expressed the need for assurances that recordings in which they might feature would not be retained permanently. Most felt the need for the deletion process to be clearly spelled out;

*"I think it'd be nice to be told it's being deleted so that you're not worried about . . . is it going to be sitting around for years on end"* [PATIENT 2]

Suggestions were made that a rolling deletion process should be utilized, with seven days being mooted as a suitable length of time, akin the usual footage management processes for public CCTV.

*"Well, you have the same sort of theory around security cameras that are up at banks. . .. I mean, the recording is there for a certain number of . . . a certain amount of time if people felt aggrieved by something and if they didn't . . . well if they didn't complain in a certain window well the recording wasn't there."* [PATIENT 1]

Whatever the time frame for retention was however needed to be made explicit to patients and furthermore needed to be adhered to. As PATIENT 8 noted;

*"If the . . . the initial . . . . what you signed on for was seven days the footage would be held for. I would be fine with that, but if it was going to be held for a month. . . . I think I'd like to know.. and I think I'd like to give permission, especially if you're going to be using as a training tool for other people involved. I think I would want to know" [PATIENT 8]*

The above comment reinforces the need for participants to know that footage would not be kept beyond a specified time but also, conversely opens up the possibility for longer term retention for training purposes. This was echoed by several participants who reinforced the need for their consent to use footage in such a way.

*"I would personally say yes to that . . . It's like educational and um yeah. . . . . . I wouldn't mind but I feel like having a notification for someone, if that were to be happening. Would just be nice. . . better" [PATIENT 6]*

PATIENT 5 was clearer on the subject of prolonged retention, again, while endorsing the use of recordings to education, training or improvement activities, they believed that not only was patient consent essential, but that patients should be offered a copy of any such footage;

*"Obviously having footage is fantastic for that and you should use it to say to someone look just here you need to do this or that, but the moment you take any recording or. . . or save any documentation, the patient either needs to be offered the opportunity to have a copy, or to just give consent." [PATIENT 5].*

The only exception to the rolling deletion or retention with consent for training, was in the case of a serious or so called "sentinel" event. Participants indicated that in the case of a significant incident being detected, footage would have to be retained and open disclosure would need to occur.

*"That would bring up . . . whole issues of I guess open disclosure or recording of complaints. . . . because you've got a whole range of . . . um . . . well evidence." [PATIENT 1]*

Participants identified scenarios such as theft, patient or staff assault, gross negligence or any other behaviors that would require "*reporting by a person witnessing it*" [PATIENT 3].

## Discussion

As was identified in our earlier work [27, 28] a lack of acceptance may prove a barrier to the effective use of these technologies, this may be the case despite the clearly demonstrated technical ability to achieve a positive outcome. HCW participants suggested that the major impediment would be patient concerns that the approach would constitute an invasion of privacy and it was this oft repeated contention that led us to investigate the matter with patients themselves.

In general, participants in the study were positive, although perhaps guardedly so, with regard to the concept of using video-based monitoring systems within the patient zone for hand hygiene compliance monitoring. These views align with the findings of several studies, notably the work of O'Donnell, Kamlin [29] for example who found that none of the parents involved in their study expressed any concerns about the video recording of neonatal resuscitations and that only 2 out of 165 declined to provide prospective consent for recordings.

Our interviews with consumers did not lend support to the notion that patients would object to the use of a within zone VMS. Most had a largely positive response to the concept,

provided, and this would seem to be a significant component with regard to acceptability, that they understood the functioning and purpose of the system, gave consent and that certain technical and operational criteria were met.

The importance of consumer engagement in their healthcare experience has been increasingly recognised, with this now enshrined in the Australian National Healthcare Standards—Standard 2: Partnering with Consumers [30]. A partnership, according to the standard, means;

> *"Providing care that is respectful; sharing information in an ongoing way; working with patients, carers and families to make decisions and plan care; and supporting and encouraging patients in their own care" [30]*

In relation to the use of a VMS, this meant that consumers wanted to know and understand the functioning of the system as well as how this approach would be of benefit to them and to others. A dichotomy existed for study participants; while, following explanation, most could identify the clear benefits, of the approach, there was also a somewhat negative emotional response. The explanation of the construct of acceptability by Parry, Pino [31] encapsulates the issue;

> *"Acceptability is a judgement based on the reasonable anticipation that involvement in a study will not cause harm to the participants, that their autonomy will be respected, and that the possible burdens associated with taking part will be outweighed by the anticipated worth" (p. 1272).*

The study participants engaged in the discussion of whether the "risks" associated with the use of video-based monitoring such as the perceived invasion of privacy outweighed benefits of efficient and accurate data collection and the subsequent potential to enhance the quality of safety of their care. While participants came out on the side of safety trumping the other risks it was also a position which was tempered by a variety of personal beliefs and experiences. This is a common trade off, for example Le Bris, Mazille-Orfanos [32] found that consumers and healthcare worker in their study valued the use of video if it improved care. Similarly Scott, Watermeyer [33] found that participants in their study valued recordings on the basis that "they may support their potential to improve services" (p. 25). This would suggest that in general, once those involved understand the benefits of the use of a VMS, they may view its use in a more positive light.

It would appear that underlying the intellectual "knowing" relating to the clear benefits of the approach there lurks "fear". This fear has been well documented in the literature where those who experienced recording have expressed feeling of stress, judgement, humiliation, anxiety, loss of privacy and dignity and a sense of guilt and shame [31, 34–37]. It would therefore essential to engage with participants and stakeholders in the recording process prior to implementation. Not only will this augment the engagement and "buy in" but it will also help to identify and limit hitherto unexpected problems through an enhanced understanding of the point of view of those involved [38]. By preparing, informing, reassuring and listening to those subject to the use of a VMS it may be possible to avoid, or at least reduce, the negative emotions often reported or expected prior to the use of video recording processes.

As has been discussed, some of this fear can be ameliorated by the technical and methodological features. Most study participants felt that if the technical aspects of the approach such as proximity activation, camera positioning, pixilation and lack of audio recording at a minimum, combined with full disclosure as to the functioning of the system and consent and/or the ability to opt out easily and without bias, were implemented then the use of a VMS would

neither represent an invasion of patient privacy nor be unacceptable. This demonstrates both the importance of certain technical and methodological aspects on the acceptability of the use of within zone video surveillance as well as the importance of communication and understanding. It must be noted that once participants could "see" and understand how the features of the system operated via the viewing of the videos they were more likely to respond positively to it. These findings are broadly similar to those of Kelly, Blackhurst [39]. Although their system did not involve the use of video and relied on proxy measures rather than the WHO 5 moments, it did involve tracking and identification of staff, both of which could have derailed the process in terms of acceptability. However, the authors noted that the inclusion of various features and protections as well as the fact that the system was "introduced sensitively with ongoing communication" (p.5) meant that it was not merely "tolerated" but was deemed "acceptable" by participants.

Accurate hand hygiene compliance data would enable a realistic picture of HCW behaviour in the clinical setting and allow for the identification of risks to patient safety due to practice gaps. It would also allow for the targeting of change programs to address for specifically areas of need. By taking into consideration the technical and methodological features described in this work, not only can the above aims be achieved, but in a manner that is acceptable to HCWs and healthcare consumers alike.

This study examined the views of a relatively small number (8) of health care consumers, however the themes across participants were relatively the same. It is acknowledged that a larger sample may uncover additional information. Similarly, all participants were English speaking and most were from an English-speaking background. Only one self-identified as 'vulnerable' owing to disability and another identified as the parent of children with special needs. Further exploration with participants from non-English speaking backgrounds and vulnerable populations would be warranted as part of the progression of this work.

## Conclusion

Despite predictions by HCWs, patients participating in this study indicated that they found the use of a within zone VMS for auditing hand hygiene broadly acceptable provided that certain technical and operational features and 'rules' were in place. In addition, they indicated that they needed to be informed as to the function and purpose of any system including the management of footage in terms of retention and deletion. Although the majority endorsed an "opt out" approach, consent was deemed important even though only one indicated that a formal written process was required.

## Supporting information

**S1 File. Participant information and consent form.**
(DOCX)

**S2 File. Interview question guide–health care consumers.**
(DOCX)

## Author Contributions

**Conceptualization:** Ramon Z. Shaban.

**Data curation:** Katherine J. McKay.

**Formal analysis:** Katherine J. McKay.

**Funding acquisition:** Ramon Z. Shaban.

**Investigation:** Katherine J. McKay.

**Methodology:** Katherine J. McKay, Ramon Z. Shaban.

**Project administration:** Katherine J. McKay, Ramon Z. Shaban.

**Supervision:** Ramon Z. Shaban.

**Validation:** Katherine J. McKay.

**Writing – original draft:** Katherine J. McKay.

**Writing – review & editing:** Katherine J. McKay, Ramon Z. Shaban.

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
