## [Decision Letter · Decision Letter 0]

2 Sep 2022

PONE-D-22-18853Video based monitoring systems for hand hygiene compliance auditing: what do patients think?PLOS ONE

Dear Dr. McKay,

Thank you for submitting your manuscript to PLOS ONE. After careful consideration, we feel that it has merit but does not fully meet PLOS ONE’s publication criteria as it currently stands. Therefore, we invite you to submit a revised version of the manuscript that addresses the points raised during the review process.

We look forward to receiving your revised manuscript.

Kind regards,

Mona Nabulsi, MD, MS

Academic Editor

PLOS ONE

Journal Requirements:

3. We note that table 1 includes an image of a patient in the study. 

5. Please ensure that you include a title page within your main document. You should list all authors and all affiliations as per our author instructions and clearly indicate the corresponding author.

6. Please amend your manuscript to include your abstract after the title page.

7. Please include your table as part of your main manuscript and remove the individual files. Please note that supplementary tables (should remain/ be uploaded) as separate "supporting information" files.

Reviewers' comments:

Reviewer's Responses to Questions

**Comments to the Author**

1. Is the manuscript technically sound, and do the data support the conclusions?

Reviewer #1: Partly

Reviewer #2: Yes

2. Has the statistical analysis been performed appropriately and rigorously? 

Reviewer #1: Yes

Reviewer #2: N/A

3. Have the authors made all data underlying the findings in their manuscript fully available?

Reviewer #1: Yes

Reviewer #2: Yes

4. Is the manuscript presented in an intelligible fashion and written in standard English?

Reviewer #1: Yes

Reviewer #2: No

5. Review Comments to the Author

Reviewer #1: > The article addresses a key important topic, yet the conducted research and the outcome are really shallow.

> Major concerns:

> - Only 8 patients participated in the study, and even the question guide was modified during the study period. This work is rather a pilot than real research. It can be a good first step to start a survey with 8 in-depth interviews to highlight critical points on acceptance of video monitoring. But the work should be continued; e.g. a short questionnaire should be designed that contains questions about the previously identified critical point. A conclusion cannot be drawn from the answers of 8 patients.

> - The recruitment was via email. The paper does not mention how many emails were sent out. If much more than 8 then the probability of selection bias is quite large.

> - The title claims that the survey was about using video monitoring to measure hand hygiene compliance. In the later part, patients start to talk about the advantages of video monitoring in general. They would accept the use of the system because it can monitoring of falls; facilitate family communication; can be useful when a patients present sign of abuse; or when the patient can be abusive because of drugs, the recording can be a piece of evidence, it can prevent theft or stop healthcare-workers to do something that they should not do. As patients conclude, the acceptance of the system depends on whether the benefits outweigh the risks. If the paper summarizes ALL benefits that a video-monitoring system can provide, then it cannot be concluded that "video monitoring system for auditing hand hygiene broadly accepted". In that case, the right conclusion would be that this theoretical super-video system is broadly accepted.

> Minor remarks:

> - At the abstract, before the 'Conclusion' part there is an empty row.

> - The abstract uses the abbreviation VMS without explaining it

> - The words like "yeah", "well", "um...", "a...", "you know" can be omitted from the quotes.

> - In Figure 1, the gray and orange are quite similar when it is black&white printed.

Reviewer #2: Dear authors,

Greetings!

Although the manuscript is well documented it needs some clarification and corrections. Please specify the design: what do you mean by "pragmatic naturalistic framework"?

Dear authors,

Greetings!

Although the manuscript is well documented it needs some clarification and corrections. Please specify the design: what do you mean by "pragmatic naturalistic framework"?

6. PLOS authors have the option to publish the peer review history of their article (what does this mean?). If published, this will include your full peer review and any attached files.

Reviewer #1: No

Reviewer #2: **Yes: **Abdolghani Abdollahimohammad

---

## [Editor Report · Decision Letter 1]

21 Oct 2022

PONE-D-22-18853R1Video based monitoring systems for hand hygiene compliance auditing: what do patients think?PLOS ONE

Dear Dr. McKay,

Thank you for submitting your manuscript to PLOS ONE. After careful consideration, we feel that it has merit but does not fully meet PLOS ONE’s publication criteria as it currently stands. Therefore, we invite you to submit a revised version of the manuscript that addresses the points raised during the review process.

We look forward to receiving your revised manuscript.

Kind regards,

Mona Nabulsi, MD, MS

Academic Editor

PLOS ONE

Journal Requirements:

Additional Editor Comments:- The sample size is very small (8 participants) which may suggest that saturation may not have been achieved. The study therefore is more of a 'PILOT' study the results of which need to be confirmed in a larger one. Please elaborate more on this limitation in the discussion. - Kindly use 'Median' instead of 'Mean" for the participants' age (small sample size).- There are several places in the manuscript with words missing letter or written wrongly (e.g. *concent *instead of *consent; * desientitised; emphysised; etc..). there are also a couple of sentences that are incomplete. Please review all text carefully and correct these words/sentences.

---

## [Author Response · Author response to Decision Letter 1]

26 Oct 2022

Please see the atttached response to reviewers document as well as the 2 x response letters

---

## [Editor Report · Decision Letter 2]

3 Feb 2023

Video based monitoring systems for hand hygiene compliance auditing: what do patients think?

PONE-D-22-18853R2

Dear Dr. McKay,

We’re pleased to inform you that your manuscript has been judged scientifically suitable for publication and will be formally accepted for publication once it meets all outstanding technical requirements.

Kind regards,

Claire Seungeun Lee

Academic Editor

PLOS ONE

---

## [Editor Report · Acceptance letter]

27 Feb 2023

PONE-D-22-18853R2 

Video based monitoring systems for hand hygiene compliance auditing: what do patients think? 

Dear Dr. McKay:

I'm pleased to inform you that your manuscript has been deemed suitable for publication in PLOS ONE. Congratulations! Your manuscript is now with our production department. 

Kind regards, 

on behalf of

Dr. Claire Seungeun Lee 

Academic Editor

PLOS ONE